# BMJ Open | Uptake of the multi-arm multi-stage (MAMS) adaptive platform approach: a trial-registry review of late-phase randomised clinical trials

Nurulamin M Noor ,[1] Sharon B Love,[1] Talia Isaacs,[2] Richard Kaplan,[1] Mahesh K B Parmar ,[1] Matthew R Sydes [1]

¹MRC Clinical Trials Unit at UCL, London, UK
²Institute of Education, University College London, London, UK

**Correspondence to**
Dr Nurulamin M Noor;
noorn@doctors.org.uk

## ABSTRACT

**Background** For medical conditions with numerous interventions worthy of investigation, there are many advantages of a multi-arm multi-stage (MAMS) platform trial approach. However, there is currently limited knowledge on uptake of the MAMS design, especially in the late-phase setting. We sought to examine uptake and characteristics of late-phase MAMS platform trials, to enable better planning for teams considering future use of this approach.

**Design** We examined uptake of registered, late-phase MAMS platforms in the EU clinical trials register, Australian New Zealand Clinical Trials Registry, International Standard Randomised Controlled Trial Number registry, Pan African Clinical Trials Registry, WHO International Clinical Trial Registry Platform and databases: PubMed, Medline, Cochrane Library, Global Health Library and EMBASE. Searching was performed and review data frozen on 1 April 2021. MAMS platforms were defined as requiring two or more comparison arms, with two or more trial stages, with an interim analysis allowing for stopping of recruitment to arms and typically the ability to add new intervention arms.

**Results** 62 late-phase clinical trials using an MAMS approach were included. Overall, the number of late-phase trials using the MAMS design has been increasing since 2001 and been accelerated by COVID-19. The majority of current MAMS platforms were either targeting infectious diseases (52%) or cancers (29%) and all identified trials were for treatment interventions. 89% (55/62) of MAMS platforms were evaluating medications, with 45% (28/62) of the MAMS platforms having at least one or more repurposed medication as a comparison arm.

**Conclusions** Historically, late-phase trials have adhered to long-established standard (two-arm) designs. However, the number of late-phase MAMS platform trials is increasing, across a range of different disease areas. This study highlights the potential scope of MAMS platform trials and may assist research teams considering use of this approach in the late-phase randomised clinical trial setting.

**PROSPERO registration number** CRD42019153910.

## INTRODUCTION

Randomised clinical trials (RCTs) are the gold standard for investigating healthcare interventions. Late-phase RCTs in particular play an important role in shaping health policy, guidelines and informing clinical practice to ensure better outcomes for patients and the public.[1] Historically, for most medical conditions, there have been few therapeutic options and long lag-times for developing new treatments. Thus, traditional two-arm, parallel-group RCTs were appropriate to perform in many instances.[2] However, nowadays there are multiple potential interventions available for testing in many medical

### Strengths and limitations of this study

► This study builds on previous publications reporting on uptake of adaptive or platform trials in general, by reporting the uptake and characteristics of multi-arm multi-stage (MAMS) platform late-phase randomised clinical trials (RCTs) over the past 20 years.

► The search strategy used did not include phase 1 or phase 2-only MAMS platform trials, however, this was because of the quite different considerations for trial teams designing and conducting early-phase exploratory compared with late-phase confirmatory RCTs.

► By conducting an extensive trial registry review and including all the most recent trial registrations, this study reports a much wider uptake of the MAMS platform design than previously suggested, across a large number of countries and different disease areas.

► It is important to note that adaptive trials by their nature are fast-moving area and there will have been further trial registrations since the April 2021 coverage in this manuscript. The information presented about included trials was accurate and up to date at the time of writing.

► We believe that reporting on the uptake and characteristics across these currently registered MAMS platform trials, enables detailed consideration and better planning for teams considering future use of these efficient designs.

conditions, and with patients, clinicians and healthcare systems demanding faster answers to clinical questions. It is clearly neither efficient nor expedient to continue with this approach of multiple, separate, individual two-arm RCTs. In addition, there are many more aspects of treatments to evaluate than ever before including: doses, durations and combinations of interventions.

Despite the rapidly rising number of registered clinical trials being performed around the world,[3] there are growing concerns about the length of time and increasing costs of performing late-phase RCTs.[4] While most late-phase RCTs do reach a conclusion, only 30%–40% are able to demonstrate efficacy of an intervention, sometimes referred to as 'positive' trials.[5 6] We would caution against the use of use of terms such as 'positive' or 'negative' and highlight the growing literature suggesting against use of these terms to describe RCT findings.[7 8] Nevertheless, these results do highlight that there is a pressing need to speed up the evaluation process in late-phase RCTs. In line with this, there has been a movement towards 'faster, better, more efficient' trials.[9] Indeed, it is now well recognised that for conditions for which there are multiple interventions worthy of investigation, consideration should be given to using multiarm trial designs.[10]

A range of different trial designs have been proposed which have been classified by some under the umbrella term of complex, innovative designs.[11] The adaptive platform trial (APT) approach using a multi-arm multi-stage (MAMS) protocol is one such design and offers many solutions to the above-described problems of traditional trial designs.[12] The MAMS platform trial typically uses a single master protocol to allow multiple primary research questions to be answered.[13] Multiple interventions can be assessed in parallel and the shared control arm allows immediate efficiency saving by reducing the overall recruitment needed. Once participants are enrolled, information accumulates during the course of the trial and adaptations can be made including whether to add in a new intervention arm or stop recruitment to an ongoing intervention arm.[14] These adaptations are typically made following interim analysis and according to predefined criteria,[15] to ensure they preserve validity and integrity of the trial. Of note, many of the MAMS-type adaptations could also be achieved using a Bayesian framework.[16] Crucially, these adaptations are made via amendment to a trial protocol rather than having to register, setup and conduct a separate, new trial entirely.[17]

To date, the uptake of adaptive designs has been lower for late-phase RCTs,[18] compared with early-phase RCTs—where they have been used with great success.[19] There are likely many reasons for lower uptake in late-phase settings, including: greater reliance on findings from late-phase confirmatory trials for regulatory purposes including marketing authorisation of medications, as well as more limited knowledge and lack of formal guidance about how best to apply these newer designs to the late-phase setting. However, given the high costs of late-phase trial programmes,[4] and the significant commercial impact

from failure to demonstrate efficacy,[20] the 'early get out' offered by stopping recruitment to interventions showing lack of benefit, or harm, makes the MAMS design potentially applicable across many different disease areas in the late-phase setting.[21]

Previous studies have focused on the uptake of adaptive trials in general,[19 22] across both early and late-phase settings,[23] or platform trials in general,[24] but without specific focus on the MAMS approach. Additionally due to the varied methods used, including restricting searches to trial publications,[25] many ongoing MAMS platform trials, including recently registered trials have not historically been captured. Despite anecdotal reports for increasing use of the MAMS platform design in late-phase settings, there is currently, no study documenting how extensive this uptake has been. Accordingly, we present a trial registry-based review to examine the worldwide uptake of APTs using MAMS protocols in the late-phase RCT setting.

## METHODS

We conducted a systematic search for APTs using MAMS protocols in late-phase trial settings (online supplemental table S1). In particular, given the knowledge and awareness that many MAMS platform trials would be unpublished, and the recognised importance of registering clinical trials, this review focused beyond trial publications only and sought to extract data from global trial registries (online supplemental table S1). This enabled us to identify many further registered but unpublished trials.

All the MAMS platform trials initially identified in the published literature reported an international clinical trial registry number and could subsequently be linked to a trial registration. All late-phase MAMS platform trials included in this review were registered on a trial registry by 1 April 2021, which was the final day of identification and data collection. This was the date used to lock the data for this manuscript, including both registration of a trial and information about ongoing trial progress. Information about trials was obtained from data linked to trial registry numbers, including registry entries, publications (if present), and review of publicly available documents such as trial protocols and statistical analysis plans.

An operational definition of a late-phase MAMS platform trial guided our search and eligibility criteria (box 1). Notably, given the differences and separate considerations for different APT approaches, we limited the search to the MAMS platform design and excluded other APTs, which did not otherwise meet the definition used in this study (box 1). In addition, we excluded early-phase trials, systematic reviews, commentaries, editorials, statistical methods or economic discussion manuscripts. We recorded key characteristics about MAMS platform trials including the year of registration, disease area under investigation, details about the leading organisation including the country within which they were based,

> **Box 1    Definition for late-phase adaptive platform trial using a multi-arm multi-stage (MAMS) protocol approach**
>
> **Late-phase, randomised-controlled trial using MAMS protocol definition**
> ► Late-phase defined as a phase 3, seamless phase 2/3 or a phase 2 trial which intended for phase 3 expansion at the outset even if this expansion does not ultimately occur.
> ► Randomisation of participants.
> ► Multiarm defined as two or more actual or intended comparison interventions. There is typically the intention to add in new intervention arms into the platform, but this would not be considered mandatory and the addition of new intervention arms may not ultimately take place.
> ► Multistage defined as two or more actual or intended stages with an interim analysis in between stages, with the ability to stop recruitment to intervention arms following interim analysis.

number and location of recruiting countries, research phase (planned seamless phase 2/3, phase 3), trial duration, estimated sample size, number and detail of intervention arms, detail about control arms, as well as nomenclature for trials and use of the MAMS term. We used the Preferred Reporting Items for Systematic Reviews and Meta-Analyses checklist as a guide for reporting of this study and prospectively registered this literature and registry-based review on PROSPERO (CRD42019153910).

### Patient and public involvement
No patient involved.

### RESULTS
Our search strategy (figure 1) retrieved 399 results from which 62 trials were identified which met the definition for a late-phase APT using an MAMS master protocol design (box 1).

Fifty per cent (31/62) of the trials used the MAMS term specifically (table 1). The 50% of trials which did not specifically use the MAMS term, instead used more generic terms such as adaptive platform, master protocol or platform trial. Of note, for the 50% of trials which used the MAMS term, this was more commonly noted if the trial was being led from the UK with 61% (19/31) of the trials using this term, or if being led from European countries (19%, 6/31). Additionally, all trials which self-identified as late-phase MAMS platform trials were being co-ordinated by a non-commercial trial organisation (100%, 31/31) (online supplemental table S2).

There was a pattern of gradual uptake of MAMS platform protocols over time with a rapid acceleration in more recent years, including a greater use of the MAMS approach across different disease areas and different countries (figure 2). After the first registered late-phase MAMS platform trial in 2001, the design was predominantly used in trials targeting oncology and infectious diseases. However, there has been subsequent greater adoption across different disease areas, widening each

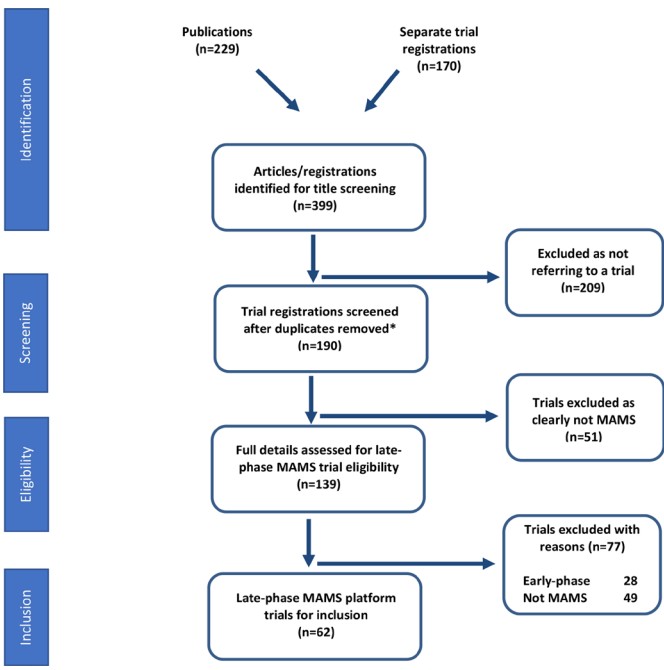

**Figure 1** Search strategy and inclusion of late-phase multi-arm multi-stage (MAMS) platform trials. This flow diagram represents the search criteria and results for late-phase MAMS platform trials identified and eligible for inclusion in this review. The date of lock for new trial registrations was 1 April 2021.

year until 2020. Indeed, since 2020, there has been a rapid uptake in the use of an MAMS platform design, accelerated by COVID-19, with 28 new trial registrations in 2020 alone. (online supplemental figure S1). In terms of publication of trial results, most of the identified late-phase MAMS platform trials in this manuscript (76%, 47/62) had not published any primary trial results at the time of data-lock for this manuscript.

In terms of leading organisation, 89% (55/62) of the late-phase MAMS platform trials identified were co-ordinated by a non-commercial organisation (table 1). In addition, although the MAMS design was found to be increasingly used around the world, 41% (26/62) of these late-phase trials were coordinated from the UK and 32% (20/62) from the USA. In terms of type of late-phase design used, 53% (33/62) of late-phase trials used a seamless phase 2/3 approach from the outset, with the remainder being phase 3 clinical trials. It is important to note that some trials were registered as seamless phase 2/3 trials with the possibility for phase 3 expansion—but did not eventually proceed to phase 3.

It is clear that trials using the MAMS design were mostly coordinated from the USA and UK initially, but that subsequently there has been greater adoption across an increasingly large number of countries including from high-income countries (HICs) and lower-income and middle-income countries (LMICs). There were 13 countries from which organisations were based and leading late-phase MAMS protocols overall and this had increased from 8 prior to the COVID-19 pandemic (figure 3A). The

**Table 1** Characteristics of included trials using an MAMS protocol approach

| Category | No of MAMS trials | Percentage of MAMS trials |
|---|---|---|
| Disease area | | |
| Infection | 32 | 52 |
| Cancer | 18 | 29 |
| Neurology | 4 | 6 |
| Mental health | 2 | 3 |
| Diabetes | 2 | 3 |
| Dermatology | 1 | 2 |
| Haematology | 1 | 2 |
| Inflammation | 1 | 2 |
| Surgery | 1 | 2 |
| MAMS term used | | |
| Yes | 31 | 50 |
| No | 31 | 50 |
| Phase of trial | | |
| Seamless phase 2/3 | 33 | 53 |
| Phase 3 | 29 | 47 |
| Organisation leading trial | | |
| Non-commercial | 55 | 89 |
| Commercial | 7 | 11 |
| Country of organisation leading trial | | |
| UK | 26 | 41 |
| USA | 20 | 32 |
| France | 5 | 8 |
| Switzerland | 2 | 3 |
| Australia | 1 | 2 |
| Austria | 1 | 2 |
| Brazil | 1 | 2 |
| Canada | 1 | 2 |
| Denmark | 1 | 2 |
| Netherlands | 1 | 2 |
| Pakistan | 1 | 2 |
| Singapore | 1 | 2 |
| Spain | 1 | 2 |

MAMS, multi-arm multi-stage.

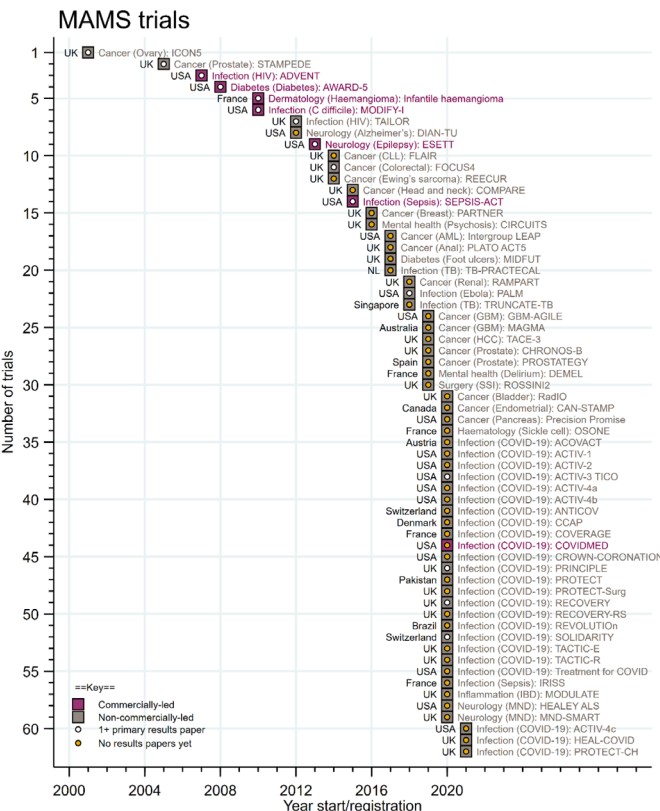

**Figure 2** Increasing uptake of late-phase trials using the multi-arm multi-stage (MAMS) platform approach over time. Hosepipe plot demonstrating increasing uptake of MAMS platform for late-phase clinical trials over time, accelerated by uptake during the COVID-19 pandemic. The date of lock for new trial registrations was 1 April 2021.

number of countries recruiting to and contributing to MAMS protocols had also more than doubled from 36 prior to the COVID-19 pandemic to 75. A large number of these recruiting countries included participation of sites from LMICs even prior to the pandemic period, typically to answer research questions for infectious diseases such as tuberculosis and Ebola (figure 3B).

The anticipated initial maximum sample size was estimated and demonstrated that generally late-phase MAMS platforms are large trials, typically expected to recruit

hundreds or even thousands of participants (online supplemental figure S2). There was no apparent association of original anticipated sample size with either the number of comparison arms in the initial MAMS platform, or with the number of arms later added on. However, the MAMS platform protocols registered for COVID-19 appeared to have greater estimated maximum sample sizes at the outset compared with non-COVID-19 trials. Of note, two of the MAMS platforms, RECOVERY initially starting in the UK and the WHO SOLIDARITY platform trial taking place across many countries around the world, specified no original sample size—instead stating that they would aim to recruit as many participants as possible to allow research questions to be answered. In both these trials, sample sizes were subsequently estimated based on interim analyses during the course of the trials.

The two most common areas for investigation using a late-phase MAMS protocol were infectious diseases (52%, 32/62) and cancer (29%, 18/62), with the infectious disease MAMS platform trials predominantly driven by those assessing potential therapeutics for COVID-19 (online supplemental table S2). All the MAMS platform protocols identified were trials of treatment interventions and 93% (58/62) of these focused on medical

**A**

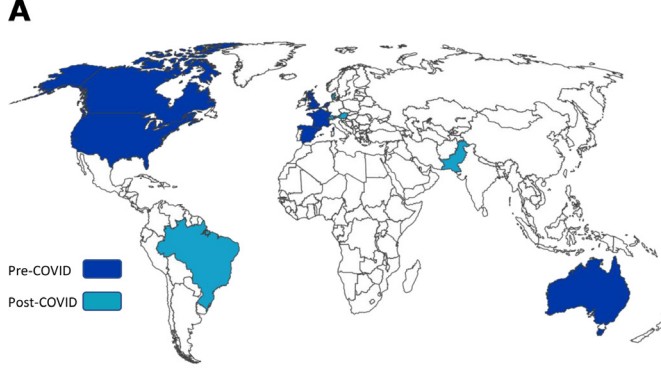

Pre-COVID ▮
Post-COVID ▮

**B**

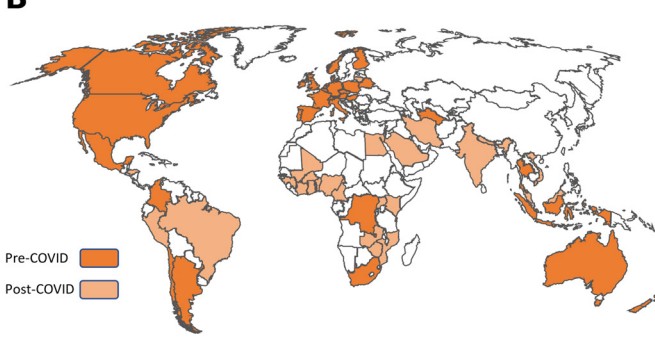

Pre-COVID ▮
Post-COVID ▮

**Figure 3** Countries involved in late-phase APTs using the MAMS protocol approach. (A) country where organisations based around the world which were leading late-phase MAMS platform trials in the pre-COVID-19 era and since the COVID-19 pandemic. (B) countries around the world recruiting to late-phase MAMS platform trials in the pre-COVID-19 era and since the COVID-19 pandemic. APT, adaptive platform trial; MAMS, multi-arm multi-stage.

treatment arms only. Indeed, the vast majority of the trials assessing medical treatments within an MAMS platform were focused on drug medications (95%, 55/58), and comparatively fewer (5%, 3/58) on interventions such as radiotherapy, oxygen or respiratory ventilation strategies (online supplemental table S3). Five per cent (3/62) of the total, identified MAMS platform trials contained surgical intervention arms and to date there was only one trial assessing psychiatric interventions. Given the growing academic interest on using repurposed medications for RCTs, we noted that 45% (28/62) of the MAMS platforms had at least one or more repurposed medication as a comparison arm, and indeed 35% (22/62) of the trials identified were studying only repurposed medications (table 2).

As highlighted above, the ability to add in new intervention arms is often regarded as a major efficiency advantage of MAMS trials. Despite 90% (56/62) of late-phase MAMS protocols prespecifying the intention or ability to add in new intervention arms into the platform trial at a later timepoint, to date, only 11% (7/62) of the trials included have actually added new intervention arms into an ongoing protocol (online supplemental table S4). Although this finding should be considered in the context that most of the late-phase MAMS trials being examined (online supplemental table S5), have been registered in

**Table 2** Characteristics of interventions and control groups for late-phase adaptive platform trial using an MAMS protocol approach

| Category | No of MAMS trials | Percentage of MAMS trials |
| --- | --- | --- |
| Type of interventions | | |
| Medical treatments only | 58 | 93 |
| Surgical treatments only | 1 | 2 |
| Medical and surgical treatments | 2 | 3 |
| Psychiatric treatments only | 1 | 2 |
| Repurposed medications | | |
| None | 29 | 47 |
| All repurposed medications | 22 | 35 |
| Some repurposed medications | 6 | 10 |
| Not applicable | 5 | 8 |
| Doses of intervention being compared | | |
| No | 57 | 92 |
| Yes | 5 | 8 |
| Durations of intervention(s) being compared | | |
| No | 59 | 95 |
| Yes | 3 | 5 |
| Plan to add in intervention arm(s) | | |
| Yes | 56 | 90 |
| No | 6 | 10 |
| Added in intervention arm(s) | | |
| No | 55 | 89 |
| Yes | 7 | 11 |
| No of intervention comparison arms at outset | | |
| 1 | 10 | 16 |
| 2 | 20 | 32 |
| 3 | 20 | 32 |
| 4 | 8 | 13 |
| 5 | 1 | 2 |
| 6 | 0 | 0 |
| 7 | 2 | 3 |
| 8 | 1 | 2 |
| No of intervention comparison arms assessed to date | | |
| 1 | 8 | 13 |
| 2 | 17 | 28 |
| 3 | 21 | 34 |
| 4 | 10 | 16 |
| 5 | 1 | 2 |
| 6 | 0 | 0 |
| 7 | 1 | 2 |
| 8 | 1 | 2 |

Continued

| Category | No of MAMS trials | Percentage of MAMS trials |
|---|---|---|
| 9 | 0 | 0 |
| 10 | 2 | 3 |
| Control arm(s) in the platform | | |
| No control group arm | 4 | 6 |
| 1 control arm—standard of care arm only | 38 | 61 |
| 1 control arm—placebo arm only | 17 | 27 |
| 2 control arms—both placebo and standard of care arms | 3 | 5% |

MAMS, multi-arm multi-stage.

last few years and still waiting for maturity, with regard trial recruitment and conduct (figure 2).

## DISCUSSION

The MAMS platform protocol approach offers considerable advantages for clinical trial efficiency compared with more traditional designs. This is the first work, to date, examining uptake of the MAMS approach in the late-phase trial setting and these findings should enable trial teams to better prepare, plan and deliver future MAMS platform protocols.

### Nomenclature of MAMS platform protocols

It has been recognised that APTs including those using an MAMS approach are typically delivered from a single, master protocol.[26] Although there have been attempts to clarify and harmonise terms,[27 28] it is clear that there are many mixed definitions and heterogeneous reporting in the clinical trials literature. Given that only 50% (31/62) of the MAMS platform trials identified used the MAMS term specifically, does illustrate that the MAMS terminology has not been universally adopted. The reasons for this are not entirely clear but may suggest a need for more knowledge-exchange and international collaboration from teams leading late-phase MAMS platform protocols. Regardless of the exact terminology used, these trials were all deemed to be of a similar nature of design, by virtue of meeting the definition outlined in box 1. As a future research priority, we support the harmonisation of terms across the clinical trials field and highlight the ongoing and commendable work from initiatives such as the European Union patient centric clinical trial platforms in this regard (www.eu-pearl.eu).

### Delivering late-phase MAMS platform protocols

There are likely to be multiple factors influencing the pattern of uptake of MAMS platform protocols. One key initial factor is likely to have been previous limited awareness of this trial design. The first practically implemented late-phase MAMS platform trial was the ICON 5 phase 3 trial in ovarian cancer, registered in 2001.[29] Recruitment occurred much more rapidly than anticipated and recruitment to all the intervention arms was stopped at the single, planned, formal interim analysis. The ICON5 trial demonstrated the merits of obtaining faster answers to allow stopping of recruitment to insufficiently interesting intervention arms.[13] However, as this trial did not proceed to a subsequent stage, it was unable to demonstrate some of the many later efficiencies of an MAMS design.[29]

Subsequently, the STAMPEDE seamless phase 2/3 trial in prostate cancer, launched in 2005 and many of the features of this trial have come to define expectations and understanding of the MAMS platform protocol.[30] STAMPEDE initially had five comparison arms compared against standard of care and is currently the longest running MAMS platform trial in the world.[31] Following these two initial applications, uptake of the MAMS design in late-phase trials was initially slow. However, we have shown that there has been a rapid increase in recent years, with an undoubted acceleration in uptake to answer research questions for the COVID-19 pandemic and obtain the fast answers required in a pandemic setting.[32 33] It is important to note though that although the MAMS platform has been accelerated by application to COVID-19, the vast majority of clinical trials conducted for COVID-19 did not use efficient trial designs such as the MAMS approach.[34] In fact, the majority of COVID-19 trials consisted of multiple, separate, individual and competing registrations, often recruiting small numbers of patients, accordingly being underpowered and ultimately failing to answer research questions for the population that they aimed to serve.[35] This illustrates the importance of increasing both knowledge and uptake for more efficient trial designs such as the MAMS platform,[36] in order to provide faster answers for patients and the public.[32]

We have demonstrated that 89% of late-phase MAMS trials have been led from non-commercial organisations such as universities, hospitals and academic clinical trials units. This is likely helped by the fact that organisations which are not bound to any single treatment or intervention, are able to run platforms and engage with collaborators to contribute a number of different interventions into the MAMS platform.

Despite the finding that MAMS platforms are typically led from non-commercial organisations, we believe that it is important to highlight the multiple sources of funding required to setup and deliver a late-phase MAMS platform protocols, often with support from many commercial partners. Indeed, without this collaboration and support, it would perhaps be impossible to deliver late-phase MAMS platform trials in most disease areas. Increasingly it is clear that, whether being coordinated and sponsored from commercial or non-commercial organisations, there are important practical considerations for trial teams and researchers, designing and delivering a late-phase MAMS

trial.[31 37 38] Indeed some of the many statistical, practical, and regulatory challenges may have contributed to the relatively low number of late-phase MAMS platform trials that have added new intervention comparison arms to date.[39–42] With regard statistical considerations, the question of whether type one error control needs to be strongly controlled or not, for multiple treatment comparisons in an MAMS platform—remains an ongoing debate and has been well covered in the recent trials literature.[43–46] Accordingly, we advise that planning, setup and delivery of such trials should be through collaboration with organisations who have prior experience of delivering large, complex, late-phase RCTs.

We have demonstrated that the majority of late-phase MAMS protocols have historically been led from HICs. However, with increasing recruitment and participation in MAMS platforms from around the world, there clearly needs to be a focus on widening leadership and co-ordination to LMICs. We also noted that the majority of the early MAMS trials were based in one country for recruitment. This is perhaps not surprising given the often stark differences in regulatory oversight,[47] ethical approvals, as well as research capacity and capability between countries, often resulting in difficulty setting up MAMS platforms in multiple countries.[48] Indeed in a platform trial where regular adaptations are made, this can be more challenging as amendments to the protocol have to be implemented at pace across participating sites.[30 49]

### Strengths and limitations

This study documents the uptake of the MAMS platform approach in late-phase RCTs over the past 20 years and builds on previous publications which have examined the uptake of adaptive trial designs. Prior publications have examined the uptake of adaptive trials designs in general without specific focus on any design,[19 22] without distinction between early and late-phase trials,[23 24] or by focusing only on published trial results.[25] Although many of these articles do also include a small number of MAMS platform trials—due to the methods used in these historical articles, many ongoing MAMS platform trials, including recently registered trials, were not captured. Accordingly, this study reports a much wider uptake of the MAMS platform design than previously suggested, across a large number of countries and many, different disease areas. By conducting an extensive trial registry review and including all the most recent trial registrations, we are able to clearly demonstrate how the COVID-19 pandemic has accelerated knowledge, use and familiarity with MAMS platform trial design and across the late-phase RCT setting.

One caveat of this current study is that most of the identified late-phase MAMS platform trials in this manuscript (76%, 47/62) have not published primary trial results to date (figure 2). This is likely to represent a need for time to achieve data maturity, and highlights that comparing across platforms and drawing too many conclusions at this stage may be premature. There may also be further MAMS platform trials globally which have

not been entered into a trial registry, however, we would highlight the importance and globally accepted good practice of registration of all late-phase RCTs onto trial registries. The search strategy used did not include phase 1 or phase 2-only MAMS platform trials, however, given the quite different considerations between early-phase exploratory and late-phase confirmatory RCTs, we felt it was appropriate to focus only on the late-phase trial setting. We have underscored the differences in terms and nomenclature used across the clinical trials field, and that some of the APTs identified may not have labelled themselves as an MAMS platform trial. In addition, other trials that some may consider similar in design, may not have been included as they did not meet the MAMS definition outlined in this manuscript (box 1). It is important to note that this is a fast-moving area and there will have been further trial registrations since the April 2021 coverage in this manuscript. The information presented about included trials was accurate and up to date at the time of writing. However, MAMS trials given their adaptive nature, will likely have been further modified, including changes to the number of intervention arms, locations of recruitment and/or information about trial reporting. Nevertheless, we believe that reporting on the uptake and characteristics across these currently registered MAMS platform trials, enables detailed consideration and better planning for teams considering future use of these efficient designs.

### Considerations for the future

We have highlighted many reasons that trial teams may consider using an MAMS platform approach for late-phase RCTs.[17] However, it is equally important to understand that these designs are not appropriate to use in all instances of late-phase RCTs.[50 51] Nevertheless, it is clear that the number of MAMS platform protocols used in the late-phase trial setting will continue to grow. There have been a few examples of narratives for implementation from teams which have conducted late-phase MAMS platform trials. In particular, there have been recent manuscripts, focusing on the many additional operational considerations, including; data management,[37] trial coordination,[38] as well as the overall experiences from teams developing and delivering large, late-phase MAMS platform trials.[31] We believe these experiences are very helpful to guide other teams looking to develop and deliver late-phase MAMS platform protocols, as well as providing an understanding for some of the regulatory, legislative and financial challenges that teams might encounter.[48] To date, published reports of experiences from trial teams delivering MAMS platform trials, remain limited. Going forward, we would urge for more experiences of implementation to be shared by each of the respective trial teams. As these designs become more widely deployed, future work could and should explore how best to further maximise efficiencies of an MAMS platform trial.

We are aware of multiple further MAMS trials that are currently being considered or designed for the late-phase

setting including, but not limited to, multiple sclerosis,[52] Parkinson's disease,[53] HIV.[54] We note that to date, there has been a lack of use for this design in areas such as cardiology. However, given the number of potential new interventions, large population sizes and ongoing medical needs in the field of cardiology, it is likely that there would be many benefits from using an MAMS design in the late-phase setting for cardiovascular conditions. In truth, no matter which disease area is being considered, we would highlight the importance of collaboration and these more efficient trial approaches being championed by key opinions leaders, funders, charities and patient groups in their respective disease areas.

## CONCLUSIONS

Since the first practically implemented late-phase MAMS platform trial, there was an initial slow uptake of this design. However, in more recent years, there has been a rapid increase with exponential uptake, and the MAMS design is now being used across a range of countries, disease areas and specialities. However, it is important to be aware that one size does not fit all and that each disease area, region and trial protocol will have their own considerations.

The use and knowledge of the MAMS design has undoubtedly been accelerated by applications in the COVID-19 pandemic, where they have been used with great success to get the fast answers needed in a pandemic. It is important to be aware of practical challenges to design, initiate and undertake these large-scale protocols and the need for collaboration across many partners as well as highlighting the need for more practical guidance. Once initiated though, it is clear that practical issues of late-phase MAMS protocols can be overcome and help drive substantial improvements in patient care. This analysis highlights the potential scope of MAMS platform trials across medicine and drug development programmes and may assist research teams considering use of this approach.

**Contributors** NMN, SBL, TI, RK, MP and MRS wrote and approved the manuscript. NMN, SBL, RK and MRS, designed the research. NMN performed the research search and extraction. NMN and MRS analysed the data and act as guarantors of this work.

**Funding** NMN is supported by a Medical Research Council PhD Studentship (MC_UU_171339).

**Map disclaimer** The inclusion of any map (including the depiction of any boundaries therein), or of any geographic or locational reference, does not imply the expression of any opinion whatsoever on the part of BMJ concerning the legal status of any country, territory, jurisdiction or area or of its authorities. Any such expression remains solely that of the relevant source and is not endorsed by BMJ. Maps are provided without any warranty of any kind, either express or implied.

**Competing interests** MRS reports grants and non-financial support from Astellas, grants from Clovis, grants and non-financial support from Janssen, grants and non-financial support from Novartis, grants and non-financial support from Pfizer, grants and non-financial support from Sanofi, during the conduct of the study; personal fees from Lilly Oncology, personal fees from Janssen, all outside of the submitted work.

**Patient consent for publication** Not applicable.

**Ethics approval** This study does not involve human participants.

**Provenance and peer review** Not commissioned; externally peer reviewed.

**Data availability statement** All data relevant to the study are included in the article or uploaded as online supplemental information.

**ORCID iDs**
Nurulamin M Noor http://orcid.org/0000-0003-3426-6408
Mahesh K B Parmar http://orcid.org/0000-0003-0166-1700
Matthew R Sydes http://orcid.org/0000-0002-9323-1371

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
