## [Reviewer comments · BMJ Open]

ARTICLE DETAILS

TITLE (PROVISIONAL)	Uptake of the multi-arm multi-stage (MAMS) adaptive platform approach: a trial-registry review of late-phase randomised clinical trials
AUTHORS	Noor, Nurulamin; Love, Sharon; Isaacs, Talia; Kaplan, Richard; Parmar, Mahesh; Sydes, Matthew

VERSION 1 – REVIEW

REVIEWER	Nepogodiev, Dmitri University of Birmingham, Academic Department of Surgery
REVIEW RETURNED	12-Oct-2021

GENERAL COMMENTS	Interesting review. Very effective figures. Minor comments:, only -The manuscript is quite long. It might be more impactful if the manuscript is a bit more focussed and punchy. The introduction and discussion could be significantly condensed. For example, I don't think it is necessary to discuss advantages of MAMS in detail as that is not the focus of the article.-Please add a sentence to the introduction section of abstract explicitly outlining study aim.-A PRISMA checklist is included in supplement but not mentioned in manuscript body. Perhaps worth adding a note that PRISMA guideline followed?-Please add the exact search string(s) to supplement. When were the literature searches performed (1 April?)-Were the search results reviewed by a second author?-It would be helpful to add trial registration ID +/- published references for all included trials to supplement-Table 3 "Number of intervention comparison arms at outset" - it took me a while to realise that this excludes control arms (hence trials with 1 arm). Is it worth clarifying this in the heading?- Table 3 "Number of intervention comparison arms assessed" - do you mean at the time of data collection? Not quite clear- Table 3 "Control arm(s) in the platform" - for clarity might be helpful to add subheadings '1 comparator arm' (placebo, standard of care only), '2 comparator arms' (both placebo and standard of care)-Figure 1: appreciate tricky to do traditional CONSORT diagram if including registry results, but I think this could be reworked to be clearer. Eg. unclear how many duplicates were excluded and at what stage. Boxes on right should have arrows coming off perpendicular to the downward arrows - ie so its clear at what stage the exclusion has happened-Figure S1: add to heading that this shows CUMULATIVE not annual registrations
---

	-Regarding specialties - I think this should be based on specialty relating to main treatment the patient is receiving in that episode rather than narrow target of the intervention in trial. Eg if there is a trial to prevent COVID in cardiac patients, I would classify this as cardiac trial as involves cardiac patients/clinicians, rather than labelling as an infection trial
--	---

REVIEWER	Lin, Jianchang Takeda Pharmaceutical Cambridge MA, Global Statistics
REVIEW RETURNED	21-Dec-2021

GENERAL COMMENTS	This paper provides nice reviews on the uptake of registered, late-phase multi-arm multi-stage (MAMS) platform trials. In general, this is a topic of relevance to the readers of journal. Few additional comments for author's consideration to improve the quality of paper:  • Platform trial is commonly used in the early phase trials, could authors comment on the rationale of not including them? • The type I error control consideration is important for late phase III trials. Although whether type I error control is needed depending on the study objective and family wise type I error rate (FWER) control is not necessary in many scenarios in platform trial, it will be more comprehensive to summarize such information (e.g. whether FWER is control or not) in the paper. • In page 7, Bayesian adaptive randomization is another important methods used in the adaptive platform trial (APT). It is important to cover briefly on this topic as well. The following reference discussed about the difference of MAMS and adaptive randomization in platform trials :  o J. Lin, V. Bunn. (2017) "Comparison of multi-arm multi-stage design and adaptive randomization in platform clinical trials". Contemporary Clinical Trials, Volume 54, 48-59 • On page 39, the review of platform on COVID-19 seem not comprehensive, e.g. I-SPY COVID or COMMUNITY trial are not included. And there are already good outcomes generated from those adaptive platform trials. • If possible, can authors also comment on the different regulatory agencies (e.g. FDA, EMA) feedbacks on the use of platform trial in late phase trials? • On page 20 discussion part, it could be helpful to add the operational considerations for the implementation of platform trials.
--

VERSION 1 – AUTHOR RESPONSE

Reviewer: 1

Interesting review. Very effective figures.

Thank you for your feedback and your subsequent detailed comments. Please find a point-by-point response below to each of your comments and suggestions, with tracked changes found in the revised manuscript.

1. The manuscript is quite long. It might be more impactful if the manuscript is a bit more focussed and punchy. The introduction and discussion could be significantly condensed. For example, I don't think it is necessary to discuss advantages of MAMS in detail as that is not the focus of the article.

We have abbreviated the advantages of the MAMS approach. Of note, in order to accommodate reviewer 2 suggestions for additional topics to include - we have now also included additional discussion on statistical and operational aspects including data management and trial co-ordination for late-phase MAMS platform trials. We have ensured that the article is below the maximum word count limit in accordance with the BMJ Open journal guidance for authors.

2. Please add a sentence to the introduction section of abstract explicitly outlining study aim.

Thank you for this suggestion. We have included a sentence to the introduction of the abstract and in line with the editorial team comments also included databases and registries searched in the methods section of the abstract.

3. A PRISMA checklist is included in supplement but not mentioned in manuscript body. Perhaps worth adding a note that PRISMA guideline followed?

Given that a systematic review was not being performed in this instance, a number of elements of PRISMA are not applicable. However, we agree that the principles of the checklist are helpful and we have now included mention of this checklist being completed, in the methods section of the manuscript.

4. Please add the exact search string(s) to supplement. When were the literature searches performed (1 April?)

The search strategy including databases, registries, search string, filters and limits used, as well as the date of searching has now been added to the supplement as a new Table S1, and accordingly this table has been referenced within the body of the manuscript. This is also in line with comments and suggestion by the editorial team.

5. Were the search results reviewed by a second author?

We have clarified that the search strategy was performed by a single reviewer. There was no double extraction of data, however subsequent review and verification of findings was based on expert knowledge of the subject area by co-authors. Through discussion with collaborators and colleagues with the final list, no additional late-phase MAMS platform trials were indicated as missing or inappropriately included from the list obtained.

6. It would be helpful to add trial registration ID +/- published references for all included trials to supplement

We originally did have individual trial registration identification numbers next to the name for each MAMS platform trial, however this did make the supplementary tables more difficult to read. We agree that this is useful information and we have now added in a separate supplementary Table S5 to provide a unique trial registration identification number, and name of the global trial registry used, to link to each identified late-phase MAMS platform trial examined in this article.

7. Table 3 "Number of intervention comparison arms at outset" - it took me a while to realise that this excludes control arms (hence trials with 1 arm). Is it worth clarifying this in the heading?

Thank you for this suggestion. This highlights one of the key efficiencies typically seen in a MAMS platform trial, with use of a shared control arm, against which comparisons are made by comparator/intervention arms. We have clarified in Table 3 and the description, that number of arms does indeed refer to intervention comparison arms within a MAMS platform.

8. Table 3 "Number of intervention comparison arms assessed" - do you mean at the time of data collection? Not quite clear

This is an important point that we have now clarified by stating intervention comparison arms assessed to date. We can confirm that this was at the time of data collection and locking of the dataset.

9. Table 3 "Control arm(s) in the platform" - for clarity might be helpful to add subheadings '1 comparator arm' (placebo, standard of care only), '2 comparator arms' (both placebo and standard of care)

Thank you for this helpful suggestion. We have now modified Table 3 in line with this recommendation.

10. Figure 1: appreciate tricky to do traditional CONSORT diagram if including registry results, but I think this could be reworked to be clearer. Eg. unclear how many duplicates were excluded and at what stage. Boxes on right should have arrows coming off perpendicular to the downward arrows - ie so its clear at what stage the exclusion has happened

We have clarified the number of separate/unique trial registrations. The flow diagram has also been modified as suggested to make clearer at what stage exclusions took place.

11. Figure S1: add to heading that this shows CUMULATIVE not annual registrations

Thank you for this important point and suggestion. The heading has been updated to reflect cumulative and not annual registrations.

12. Regarding specialties - I think this should be based on specialty relating to main treatment the patient is receiving in that episode rather than narrow target of the intervention in trial. Eg if there is a trial to prevent COVID in cardiac patients, I would classify this as cardiac trial as involves cardiac patients/clinicians, rather than labelling as an infection trial

We agree entirely that categorising trials into different specialities can be difficult, with no uniformly accepted classification system. Broadly speaking the category could focus on the population eligible for inclusion, on the disease area under investigation, or on the intervention(s) being assessed in the MAMS platform. We appreciate that others may have chosen to categorise in a different way with regards to speciality. In this manuscript, we have categorised based on the disease area under investigation, and based on the interventions being investigated – as we felt this would be of most interest and of most practical use to clinical trialists seeking to use the MAMS platform approach for future trials.

Reviewer: 2

This paper provides nice reviews on the uptake of registered, late-phase multi-arm multi-stage (MAMS) platform trials. In general, this is a topic of relevance to the readers of journal.

Thank you for your positive comments and for your careful consideration of this manuscript. Please find detailed responses to each of your comments below, with tracked changes found in the revised manuscript.

1. Platform trial is commonly used in the early phase trials, could authors comment on the rationale of not including them?

We thank the reviewer for this important point. Previous publications have focused on the wider use of more innovative designs in the early-phase clinical trial setting – this has included a widespread move towards more model-based designs. In contrast, there has been less use of innovative trial designs in the late-phase setting. There are likely to be multiple reasons for this and we highlight some of the reasons in this manuscript. Therefore, we felt that it was important to focus on uptake in the late-phase clinical trial setting and hope to draw lessons across these trials to support the conduct of future late-phase adaptive platform trials - using specifically a MAMS approach. We would also highlight that the MAMS platform approach was also initially designed for the late-phase setting. We have made this aim clearer in the introduction section, which is also in line with the helpful comments and suggestion from reviewer 1.

2. The type I error control consideration is important for late phase III trials. Although whether type I error control is needed depending on the study objective and family wise type I error rate (FWER) control is not necessary in many scenarios in platform trial, it will be more comprehensive to summarize such information (e.g. whether FWER is control or not) in the paper.

We entirely agree that type 1 error control is an interesting and topical area. This subject has been well covered in the recent clinical trials literature. We have now added a short section highlighting the importance of this topic and have cited four manuscripts which discuss this subject in detail (Bratton D et al. *Trials*, 2016; Proschan M et al. *Stat Med*, 2020; Parker C et al *Clin Trials*, 2020; Wason J et al. *Pharm Stat*, 2021).

3. In page 7, Bayesian adaptive randomization is another important methods used in the adaptive platform trial (APT). It is important to cover briefly on this topic as well. The following reference discussed about the difference of MAMS and adaptive randomization in platform trials :

- o J. Lin, V. Bunn. (2017) “Comparison of multi-arm multi-stage design and adaptive randomization in platform clinical trials”. *Contemporary Clinical Trials*, Volume 54, 48-59

Thank you for this point and the suggested reference. We fully agree that many of the MAMS-type adaptations could be achieved within an adaptive platform trial (APT) using a Bayesian framework. We have now added this in the discussion and have included the suggested reference J. Lin, V. Bunn. (2017) “Comparison of multi-arm multi-stage design and adaptive randomization in platform clinical trials”. *Contemporary Clinical Trials*, Volume 54, 48-59.

4. On page 39, the review of platform on COVID-19 seem not comprehensive, e.g. I-SPY COVID or COMMUNITY trial are not included. And there are already good outcomes generated from those adaptive platform trials.

We can confirm that the search and data were up-to-date at the time of collection and locking of the database. However, over time and given the adaptive nature of MAMS platform trials, we fully understand that there will be more trial registrations and adaptations within ongoing trial platforms. With regards the two specific trials highlighted, the COMMUNITY trial could not at the time be

identified in any of the global clinical trial registries using the search strategy and terms specified. We have only found press releases on this trial, without reference to clinical trial registration numbers, or links to any publications, or indeed open-access documents such as a trial protocol or statistical analysis plan. This may be because none of the broad search terms used were included in the trial registration or indeed that the trial was not registered at the time. Indeed, we already highlight in the potential limitations of our work, that there may be further MAMS platform trials globally, which have not been entered into a trial registry or not found using the broad search terms we used. However, we would highlight the importance and globally accepted good practice for registration of all late-phase RCTs onto trial registries and of the need to harmonise terminology in this field. Regarding, I-SPY COVID, this was found in the search but was strictly focusing on the phase 2 trial setting, therefore did not meet the eligibility criteria for inclusion. Although early-phase trials are clearly very important, this manuscript focuses on the application and uptake of late-phase MAMS platform trials - for the reasons already highlighted in the introduction.

5. If possible, can authors also comment on the different regulatory agencies (e.g. FDA, EMA) feedbacks on the use of platform trial in late phase trials?

Thank you for this question. This is an important but extremely large topic which we feel is beyond the scope of our article. We note that many international regulators either lack guidance on this area or still have guidance documents in the draft or planning stage with regards complex, innovative design (CID) trials. However, we fully agree that there is an important role for regulatory agencies to ensure that pathways are in place - to enable clinical trials to use a MAMS platform approach.

6. On page 20 discussion part, it could be helpful to add the operational considerations for the implementation of platform trials.

Thank you for this excellent suggestion. We wholeheartedly agree that there are many additional operational considerations for trials using a MAMS platform approach. We have now added in a short discussion on this topic and highlighted three recent papers examining operational considerations from a data management point of view, a trial management point of view, as well as experiences within the wider team when delivering these more complex and innovative MAMS platform trials in the late-phase setting (Hague D et al. *Trials*, 2019; Schiavone F et al. *Trials*, 2019; Morrell L et al. *Trials*, 2019).

VERSION 2 – REVIEW

REVIEWER	Nepogodiev, Dmitri University of Birmingham, Academic Department of Surgery
REVIEW RETURNED	13-Jan-2022

GENERAL COMMENTS	Thank you for your considered response to previous reviewer comments. I have no further comments to make.
---

REVIEWER	Lin, Jianchang Takeda Pharmaceutical Cambridge MA, Global Statistics
REVIEW RETURNED	12-Jan-2022

GENERAL COMMENTS	my questions have been addressed by authors. I have no further comments.
--